# Anticholinergic burden: First comprehensive analysis using claims data shows large variation by age and sex

Jonas Reinold[1]*, Malte Braitmaier[2], Oliver Riedel[1], Ulrike Haug[1,3]

**1** Department of Clinical Epidemiology, Leibniz Institute for Prevention Research and Epidemiology–BIPS, Bremen, Germany, **2** Department of Biometry and Data Management, Leibniz Institute for Prevention Research and Epidemiology–BIPS, Bremen, Germany, **3** Faculty of Human and Health Sciences, University of Bremen, Bremen, Germany

* reinold@leibniz-bips.de

## Abstract

### Purpose

The cumulative effect of medication inhibiting acetylcholine activity—also known as anticholinergic burden (AB)—can lead to functional and cognitive decline, falls, and death. Given that studies on the population prevalence of AB are rare, we aimed to describe it in a large and unselected population sample.

### Methods

Using the German Pharmacoepidemiological Research Database (GePaRD) with claims data from ~20% of the German population we analyzed outpatient drug dispensations in 2016. Based on the Anticholinergic Cognitive Burden (ACB) scale, we classified persons into four categories and determined the cumulative AB as continuous variable.

### Results

Among 16,470,946 persons (54% female), the prevalence of clinically relevant AB (ACB≥3) was 10% (women) and 7% (men). Below age 40 it was highest in persons ≤18 years (6% both sexes). At older ages (50–59 vs. 90–99 years), prevalence of ACB≥3 increased from 7% to 26% (men) and from 10% to 32% (women). Medication classes contributing to the cumulative AB differed by age: antihistamines, antibiotics, glucocorticoids (≤19 years), antidepressants (20–49 years), antidepressants, cardiovascular medication, antidiabetics (50–64 years), and additionally medication for urinary incontinence/overactive bladder (≥65 years). Medication dispensed by general physicians contributed most to the cumulative AB.

### Conclusion

Although a clinically relevant AB is particularly common in older persons, prevalence in younger age groups was up to 7%. Given the risks associated with AB in older persons, targeted interventions at the prescriber level are needed. Furthermore, risks associated with AB in younger persons should be explored.

**Data Availability Statement:** In Germany, use of personal data is protected by the Federal Data Protection Act and particularly the use of claims data for research is regulated by the Code of Social

Law. Researchers have to apply for a project-specific permit from the statutory health insurance providers which then need an approval from their governing authorities. The use of the data on which this publication is based was only allowed for BIPS employees within the framework of the specified project and limited to a pre-defined time span. Researchers who want to access the data on which this publication is based need to ask for new approval by the statutory health insurance providers DAK-Gesundheit (service@dak.de), die Techniker (service@tk.de), hkk Krankenkasse (info@hkk.de) and AOK Bremen/Bremerhaven (info@hb.aok.de) which upon granting approval would have to ask their respective authorities for approval. Please contact gepard@leibniz-bips.de for help with this process. The authors confirm that they had no special access privileges to the data and that other researchers will be able to access the data in the same manner as the authors by following the instructions described above.

**Funding:** The authors received no external funding for this work.

**Competing interests:** UH, OR, MB and JR are working at an independent, non-profit research institute, the Leibniz Institute for Prevention Research and Epidemiology – BIPS. Unrelated to this study, BIPS occasionally conducts studies financed by the pharmaceutical industry. Almost exclusively, these are post-authorization safety studies (PASS) requested by health authorities. The design and conduct of these studies as well as the interpretation and publication are not influenced by the pharmaceutical industry. The study presented was not funded by the pharmaceutical industry. The authors have no relevant financial or non-financial interests to disclose. Moreover, this does not alter our adherence to PLOS ONE policies on sharing data and materials.

## Introduction

Medications with anticholinergic activity (MACs) inhibit the effect of the neurotransmitter acetylcholine [1]. They are used for the treatment of diseases such as depression, psychosis, cardiovascular diseases, asthma, overactive bladder, and COPD [1]. The cumulative effect of MACs, also called anticholinergic burden (AB), has been shown to be associated with adverse health outcomes such as functional [2, 3] and cognitive decline [2, 4, 5], delirium [6, 7], falls [3, 8], and death [9, 10].

Although the majority of studies on the adverse effects of AB focused on older adults, there are studies suggesting that younger populations might also be affected. In some of those studies, AB was associated with impaired cognitive ability and real-world functioning as well as a negative impact on the outcomes of psychosocial treatment programs in patients with schizophrenia or schizoaffective disorder [11]. Notably, many of the medications contributing to the AB had indications other than psychiatric diseases [11]. Some studies showed impairment of verbal learning and/or verbal memory associated with AB in persons with schizophrenia [12–14] and major depressive disorder [15]. Furthermore, studies have shown an association between AB and delirium in pediatric intensive care patients [16] and critically ill middle-aged adults [17]. These data suggest that already in younger patients, AB might be associated with adverse effects. So far, only a single study has provided a comprehensive overview of the prevalence of AB in all age groups of a population [18]. However, in this study, age categories were defined broadly and AB prevalences were not stratified by sex within age groups.

In our study, we aimed to characterize the prevalence of AB in a large and unselected sample of the German general population and to assess the classes of medication contributing to the total cumulative AB, stratified by age and sex.

## Methods

### Data source

We used the German Pharmacoepidemiological Research Database (GePaRD), which is based on claims data from four statutory health insurance providers in Germany and currently includes information on approximately 25 million persons who have been insured with one of the participating providers since 2004 or later. Per data year, there is information on approximately 20% of the general population and all geographical regions of Germany are represented. In Germany, about 90% of the general population are covered by statutory health insurance. The health care system is characterized by uniform access to all levels of care and free choice of providers.

In addition to demographic data, GePaRD contains information on outpatient drug dispensations as well as outpatient (i.e., from general practitioners and specialists) and inpatient services and diagnoses. Information on medication includes the anatomical-therapeutic-chemical (ATC) code, the prescription and dispensation date, the specialty of the prescriber as well as the number of defined daily doses (DDDs). Diagnoses are coded according to the German modification of the International Classification of Diseases and Related Health Problems, 10th Revision (ICD-10-GM).

### Study design and study population

We conducted a cross-sectional study using data from the year 2016, the most recent data at the time of analysis, to assess the prevalence of AB. We included all persons with at least one day of insurance coverage during the observation period, i.e., between 1 January and 31 December 2016 preceded by at least 365 days of continuous insurance (pre-observation

| Assessment period for morbidities | Inclusion criteria | 2016 |
|---|---|---|
| Any time prior to date of inclusion[1] (Length of assessment period dependent on length of insurance-membership; earliest start of assessment January 1st 2004) | ≥1 day of insurance within January 1st and December 31st 2016, preceded by 365 days of continuous insurance, resident in Germany, valid information on age and sex and not hospitalized for ≥90 days | Assessment of MAC[2] use among included persons, taking into account the available (continuous) observation period in 2016. For persons with a hospitalization starting in 2016 and with a duration of ≥90 days, MAC[2] use was only assessed until the start of this hospitalization. |
| **Assessment period for medication to describe morbidities** | | |
| Within 365 days prior to date of inclusion[1] | | |

[1] A person's date of inclusion was defined as the first date between January 1st and December 31st 2016 on which all inclusion criteria were fulfilled

[2] MAC: Medication with anticholinergic activity

**Fig 1. Graphical depiction of study design.**

period). We excluded persons with a place of residence outside of Germany, without valid information on age and sex as well as persons with a hospitalization of ≥90 days, which overlapped into this person's observation period. For all included persons, the available (continuous) observation period in 2016 was used to assess the use of MAC. For persons with a hospitalization starting in 2016 and with a duration of ≥90 days, MAC use was only assessed until the start of this hospitalization (Fig 1).

We identified morbidities and treatment with medication excluding MAC using sensitive identification algorithms: The coding of morbidities was assessed any time prior to observation period (starting from 2004) through records of ≥1 ICD-10-GM inpatient or outpatient diagnoses or records of ≥1 codes of relevant operations, procedures or outpatient services as well as participation in disease management plans. This approach, i.e. taking into account all information on morbidity available for a person before 2016, aims to compensate for the fact that with secondary data, a person cannot be asked if he or she ever had a certain disease, as it would be done in a study based on primary data. Treatment with medication excluding MAC was assessed within 365 days before start of observation period (excluding start of observation period) based on records of ≥1 outpatient dispensations.

## Assessment of the anticholinergic burden

Exposure to MAC was assessed based on outpatient prescriptions dispensed during the observation period, i.e., in 2016. Treatment durations were estimated based on DDDs. In case MAC were dispensed before 1 January 2016 and the days of supply covered by this dispensation overlapped with the observation period, the DDDs overlapping with the observation period were also considered. We assumed lower DDDs for persons aged ≤18 and ≥65 years if recommended in the respective Summary of Product Characteristics. Moreover, we identified the specialty of the prescribing physician for each dispensation of MAC. To quantify the AB in individuals, we used a list of relevant MAC and a scoring system proposed by Kiesel et al. [19]. Kiesel et al. systematically reviewed published lists of MAC and corresponding scores, mainly developed in the US, UK or Australia, and adapted them to medications relevant for Germany [19]. Their categorization of AB [19] was based on the Anticholinergic Cognitive Burden (ACB) scale, which was developed by Boustani et al. to identify persons at risk for cognitive

impairment [20]. Based on this scoring system, MACs dispensed during the observation period were scored according to their anticholinergic effects: ACB score 1 (evidence from in vitro data that chemical entity has antagonist activity at muscarinic receptor), ACB score 2 (evidence from literature, prescriber's information, or expert opinion of clinical anticholinergic effect) or ACB score 3 (evidence from literature, expert opinion, or prescriber's information that medication may cause delirium) [20, 21]. Boustani et al. considered dispensation of MAC with an ACB score 2 or 3 as well as a total ACB score of 3 or higher as clinically relevant [20]. For the interpretation of this study, we defined ACB≥3 as clinically relevant and additionally considered ACB categories ACB = 0, ACB = 1, ACB = 2, and ACB≥3 in order to assess borderline AB in the study population. For our study population, the AB was calculated for each person on a daily basis during the observation period by adding up the scores of all dispensed MACs. Prevalence of morbidities, treatment with medication other than MAC, and health care utilization were stratified by the highest category of AB reached during the observation period.

We also calculated a measure which we called "cumulative AB". We calculated this additional measure because it allowed us to assess the proportion of AB attributable to a certain class of MAC (e.g., antidepressants) or to a certain physician specialty. This measure was called "cumulative burden" because it takes into account all dispensations in the observation period (i.e. in 2016). This cumulative AB was calculated as follows for each person: We first multiplied the AB score of each MAC dispensed to the person during the observation period or overlapping the observation period with the length of supply (based on DDD) and then summed up the score points of all dispensations. Subsequently, these AB scores were summed up per person to calculate the cumulative AB. For example, a person receiving 200 DDDs of metformin (ACB score 1) and 30 DDDs of tramadol (ACB score 2) during the observation period had a cumulative AB of 260 (i.e., the result of 200 x 1 + 30 x 2). This method was proposed by Campbell et al. [5]. Campbell et al. further divided the cumulative AB by the number of days in the exposure period to transfer the total AB score into a mean score per person but this additional transformation was not relevant in the context of our study [5].

## Data analysis

We calculated the period prevalence of AB for each of the four AB categories for the observation period. The prevalence was calculated as the number of persons in the respective AB category (numerator) divided by the number of included persons (denominator). Again, persons were allocated to the highest level of AB reached during the observation period.

In order to describe which proportion of the cumulative AB was attributable to a certain class of MAC (e.g., antidepressants) or physician specialty (e.g., general practitioners), the cumulative AB of a MAC class or physician specialty of each respective age and sex group was divided by the total cumulative AB in that age and sex group.

Data management and analyses were conducted using SAS 9.4 (SAS Institute Inc., Cary, NC, USA).

## Ethics and approvals

In Germany, the utilization of health insurance data for scientific research is regulated by the Code of Social Law. All involved health insurance providers as well as the German Federal Office for Social Security and the Senator for Health, Women and Consumer Protection in Bremen as their responsible authorities approved the use of GePaRD data for this study. Informed consent for studies based on claims data is required by law unless obtaining consent appears unacceptable and would bias results, which was the case in this study. According to

the Ethics Committee of the University of Bremen studies based on GePaRD are exempt from institutional review board review.

## Results

The study population included a total of 16,470,946 persons (53.6% female) with a median age of 45 years (Q1–Q3: 26–61 years) (Fig 2).

For the majority of the study population, we observed no AB during the observation period, i.e., ACB = 0 in 68.5% of men and in 61.7% of women (Table 1). Prevalence of ACB = 1 was 17.6% in men and 19.7% in women, for ACB = 2, it was 6.7% in men and 8.2% in women, while a clinically relevant AB (ACB≥3) was observed in 7.2% of men and 10.4% of women (Fig 3).

Both in men and women, the prevalence of ACB≥3 was about 6% in persons aged ≤18 years and thus higher than in persons aged 19–49 years. At older ages, the prevalence of ACB≥3 steadily increased. In men, it increased from 7.2% (50–59 years) to 11.1% (60–69 years) and 17.2% (70–79 years). The same pattern was seen in women but the prevalences were about 3–4 percentage points higher (50–59 years: 10.6%, 60–69 years: 14.8%, 70–79 years: 21.9%).

For all morbidities and medications assessed prior to start of observation period, prevalences increased with increasing ACB score (S1 Table). For example, compared to persons with lower or no ACB, persons with ACB≥3, had higher prevalences of psychiatric and behavioral, musculoskeletal as well as endocrine and metabolic diseases. They were prescribed medications from a higher number of different prescribers and had higher prevalences of cardiovascular therapy, analgesics and psychiatric medication. Moreover, persons with ACB≥3 were, on average, more frequently hospitalized, remained hospitalized for longer periods and had a higher prevalence of nursing home residency and obesity.

Persons with ACB≥3 were more frequently users of antidepressants (45.3% vs. 8.8%), antihistamines (17.7% vs. 7.1%), and antipsychotics (13.9% vs. 1.3%) (Table 2). Individuals who used medications for urinary incontinence/overactive bladder had ACB≥3 by default (13.0%), since all of these medications have an ACB score of 3.

Median total cumulative burden increased with higher age, was highest among the age group 80–94 years, and decreased slightly in age group ≥95 years (Fig 4).

The contribution of the medication classes of MAC to the total cumulative AB differed between age groups (Table 3). In persons aged ≤19 years, antihistamines and antibiotics contributed most—with about 20–24% each—to the cumulative burden, followed by glucocorticoids with about 12–13%. In females, the contribution of antidepressants to the cumulative AB was twice as high as in males (16% vs. 8%). In persons aged 20–64 years, antidepressants

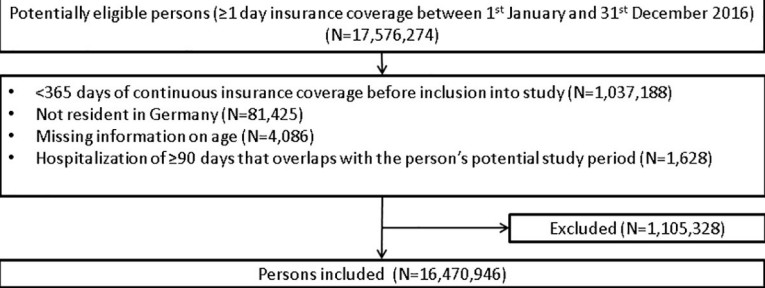

**Fig 2. Flow chart illustrating the inclusion and exclusion of persons into the study.**

**Table 1. Number and period prevalence of persons with and without anticholinergic burden measured through the Anticholinergic Cognitive Burden (ACB) scale during the observation period (2016), by sex and age.**

| | | ACB score | | | | | | | |
|---|---|---|---|---|---|---|---|---|---|
| | Total | ACB = 0 | | ACB = 1 | | ACB = 2 | | ACB ≥ 3 | |
| Sex | N[a] | N[b] | Prevalence (%) | N[b] | Prevalence (%) | N[b] | Prevalence (%) | N[b] | Prevalence (%) |
| **Men** | 7,635,507 | 5,232,064 | 68.5 | 1,342,836 | 17.6 | 507,767 | 6.7 | 552,840 | 7.2 |
| **Age** | | | | | | | | | |
| ≤ 18 | 1,361,884 | 1,069,419 | 78.5 | 185,882 | 13.6 | 30,520 | 2.2 | 76,063 | 5.6 |
| 19 to 29 | 1,128,700 | 956,617 | 84.8 | 121,521 | 10.8 | 31,648 | 2.8 | 18,914 | 1.7 |
| 30 to 39 | 1,077,901 | 862,425 | 80.0 | 138,917 | 12.9 | 44,353 | 4.1 | 32,206 | 3.0 |
| 40 to 49 | 975,152 | 715,119 | 73.3 | 158,347 | 16.2 | 55,720 | 5.7 | 45,966 | 4.7 |
| 50 to 59 | 1,210,955 | 790,562 | 65.3 | 240,566 | 19.9 | 92,866 | 7.7 | 86,961 | 7.2 |
| 60 to 69 | 853,986 | 448,660 | 52.5 | 214,327 | 25.1 | 96,021 | 11.2 | 94,978 | 11.1 |
| 70 to 79 | 697,604 | 285,943 | 41.0 | 191,496 | 27.5 | 100,439 | 14.4 | 119,726 | 17.2 |
| 80 to 89 | 293,902 | 93,090 | 31.7 | 82,108 | 27.9 | 49,783 | 16.9 | 68,921 | 23.5 |
| 90 to 99 | 35,049 | 10,119 | 28.9 | 9,557 | 27.3 | 6,353 | 18.1 | 9,020 | 25.7 |
| ≥ 100 | 374 | 110 | 29.4 | 115 | 30.7 | 64 | 17.1 | 85 | 22.7 |
| **Women** | 8,835,439 | 5,447,201 | 61.7 | 1,741,731 | 19.7 | 728,329 | 8.2 | 918,178 | 10.4 |
| **Age** | | | | | | | | | |
| ≤ 18 | 1,286,334 | 1,017,119 | 79.1 | 167,010 | 13.0 | 28,592 | 2.2 | 73,613 | 5.7 |
| 19 to 29 | 1,120,182 | 867,427 | 77.4 | 175,136 | 15.6 | 47,879 | 4.3 | 29,740 | 2.7 |
| 30 to 39 | 1,154,740 | 850,292 | 73.6 | 194,824 | 16.9 | 62,185 | 5.4 | 47,439 | 4.1 |
| 40 to 49 | 1,214,399 | 804,947 | 66.3 | 236,844 | 19.5 | 87,472 | 7.2 | 85,136 | 7.0 |
| 50 to 59 | 1,516,942 | 888,755 | 58.6 | 329,169 | 21.7 | 138,788 | 9.1 | 160,230 | 10.6 |
| 60 to 69 | 1,086,934 | 531,914 | 48.9 | 264,899 | 24.4 | 128,807 | 11.9 | 161,314 | 14.8 |
| 70 to 79 | 923,303 | 344,264 | 37.3 | 238,524 | 25.8 | 138,632 | 15.0 | 201,883 | 21.9 |
| 80 to 89 | 429,269 | 117,652 | 27.4 | 109,445 | 25.5 | 76,061 | 17.7 | 126,111 | 29.4 |
| 90 to 99 | 100,998 | 24,229 | 24.0 | 25,221 | 25.0 | 19,451 | 19.3 | 32,097 | 31.8 |
| ≥ 100 | 2,338 | 602 | 25.7 | 659 | 28.2 | 462 | 19.8 | 615 | 26.3 |

[a] Denominator of the prevalence are persons insured for ≥1 day within the observation period and with ≥1 year continuous insurance before.

[b] Numerator of the prevalence, calculated as the number of persons with ACB = 0, ACB = 1, ACB = 2, and ACB ≥ 3, respectively. Persons will be allocated in the highest level of the ACB score ever reached during the observation period.

contributed most to the cumulative AB, with proportions ranging between 25% in men aged 50–64 years to 48% in women aged 20–34 years. From age group 65–79 onwards, cardiovascular medication contributed to 24–26% of the AB in men and 21–23% in women. The proportion of diuretics increased particularly from age group 65–79 onwards and contributed to 6–19% of the cumulative AB in men and 6–17% in women. Also, the contribution of medication for urinary incontinence or overactive bladder increased with higher age to up to 14% (men aged 80–94 years). The contribution of antidiabetics to the cumulative AB was highest in men aged 50–79 years (17–19%). The contribution of medication for the treatment of respiratory diseases, gastrointestinal medications, and opioids increased slightly in persons aged ≥65 years, while the contribution of glucocorticoids to the AB decreased.

Prescriptions from general practitioners were the main contributors to the cumulative AB (Table 4). The proportion ranged between 40 and 41% in persons aged 20–34 years and increased to over 70% and more in persons aged 65 or older. In the age groups 20–49 years, prescriptions from physicians specializing in psychology and psychiatry contributed to about one fourth of the total cumulative AB. The number of different physician specialties that

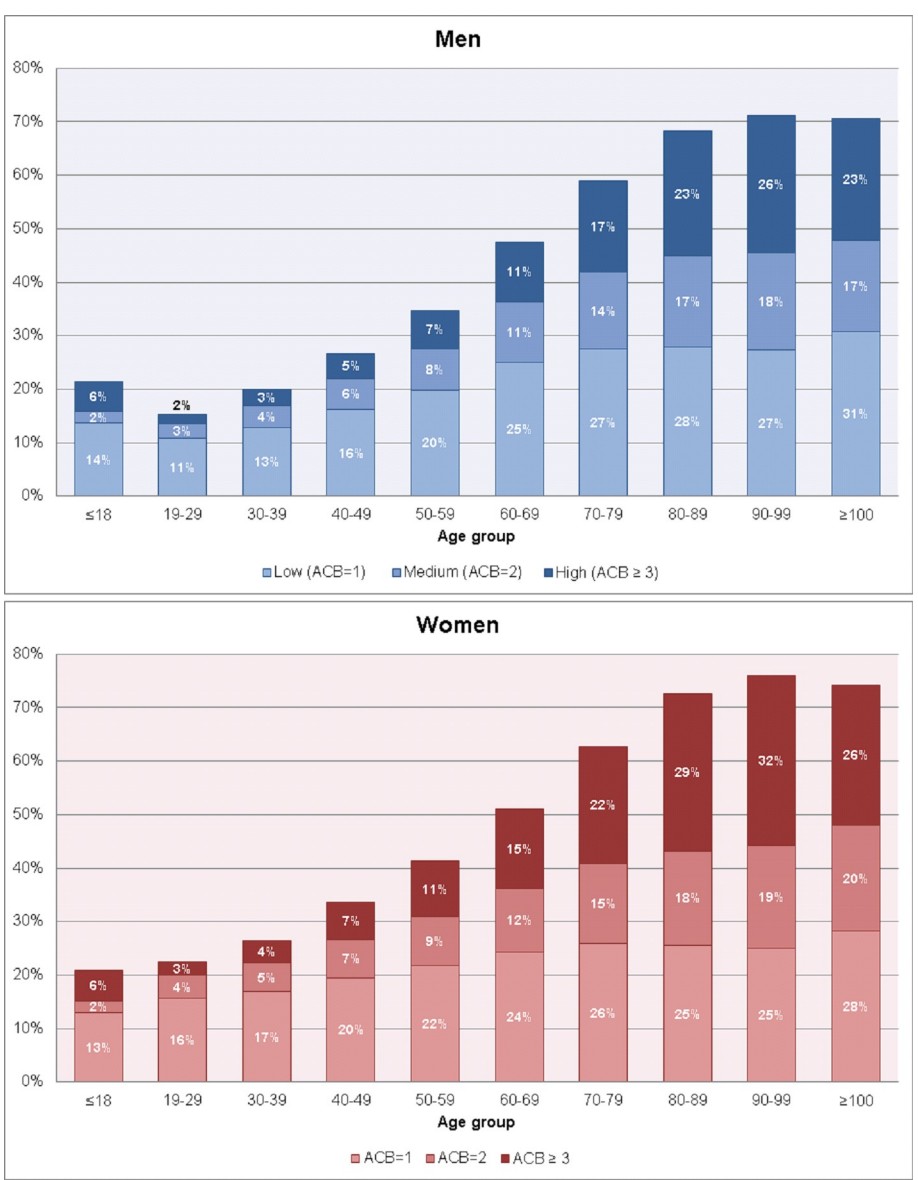

**Fig 3. Proportion of anticholinergic burden measured through the Anticholinergic Cognitive Burden (ACB) scale (2016), by sex and age.**

contributed 5% or more to the cumulative AB was five in persons aged ≤19 years, 3–4 in persons aged 20–64 years, and 2–3 in persons aged ≥65 years.

## Discussion

In our study, which included an unselected sample of 16 million persons of the German general population, about 7% of men and 10% of women had a clinically relevant AB (ACB≥3) based on prescriptions in 2016. The prevalence of ACB≥3 was higher in women than in men across all age groups and—even though increasing with age—already reached levels of 2–7% (men) and 3–11% (women) in persons younger than 60 years. The classes of medication contributing to the total cumulative AB differed greatly between sex and age groups: While antidepressants had a dominant share in age groups <60 years, their relative proportion decreased

**Table 2. Prevalence of use of medications with anticholinergic activity (MACs) in persons with anticholinergic burden measured through the anticholinergic cognitive burden (ACB) scale during the observation period (2016).**

| MAC class | ACB score[a] | | |
|---|---|---|---|
| | ACB = 1 | ACB = 2 | ACB ≥ 3 |
| | N = 3,084,567[b] | N = 1,236,096[b] | N = 1,471,018[b] |
| Antidepressants | 271,776 (8.8%) | 272,488 (22.0%) | 665,675 (45.3%) |
| Antihistamines | 218,228 (7.1%) | 63,216 (5.1%) | 260,137 (17.7%) |
| Antipsychotics | 40,943 (1.3%) | 66,412 (5.4%) | 204,098 (13.9%) |
| Benzodiazepines | 66,391 (2.2%) | 54,438 (4.4%) | 143,152 (9.7%) |
| Cardiovascular medication | 487,324 (15.8%) | 276,345 (22.4%) | 343,257 (23.3%) |
| Diuretics | 58,088 (1.9%) | 63,387 (5.1%) | 111,202 (7.6%) |
| Gastrointestinal medication | 8,208 (0.3%) | 47,898 (3.9%) | 90,185 (6.1%) |
| Opioids | 200,374 (6.5%) | 170,598 (13.8%) | 275,654 (18.7%) |
| Medication for Parkinson's disease | 32,517 (1.1%) | 31,349 (2.5%) | 91,838 (6.2%) |
| Medication for urinary incontinence/overactive bladder | 0 (0.0%) | 0 (0.0%) | 191,768 (13.0%) |
| Medication for respiratory diseases | 125,958 (4.1%) | 85,436 (6.9%) | 135,733 (9.2%) |
| Glucocorticoids | 697,422 (22.6%) | 332,459 (26.9%) | 414,559 (28.2%) |
| Tropane alkaloids | 0 (0.0%) | 0 (0.0%) | 9,131 (0.6%) |
| Immunosuppressants | 10,696 (0.3%) | 11,859 (1.0%) | 14,953 (1.0%) |
| Muscle relaxants | 123,733 (4.0%) | 44,098 (3.6%) | 99,581 (6.8%) |
| Antiemetics | 203,433 (6.6%) | 72,146 (5.8%) | 133,709 (9.1%) |
| Antibiotics | 444,014 (14.4%) | 311,271 (25.2%) | 215,548 (14.7%) |
| Antiepileptics | 16,259 (0.5%) | 29,071 (2.4%) | 58,048 (3.9%) |
| Non-opioid analgesics | 126,367 (4.1%) | 68,198 (5.5%) | 95,586 (6.5%) |
| Antidiabetics | 233,543 (7.6%) | 161,959 (13.1%) | 179,913 (12.2%) |
| Other MAC | 41,977 (1.4%) | 40,833 (3.3%) | 48,087 (3.3%) |

[a] Categorization based on the highest level of the ACB score ever reached during the observation period.

[b] Denominator is the number of included persons who had ≥1 dispensation of MAC for ≥1 day during the observation period.

among persons aged ≥60 years due to the increased prescribing of cardiovascular medication and antidiabetics with anticholinergic activity.

As of now, only one other study has assessed the prevalence of AB without limitations on age or certain patient groups. The study of Cebron Lipovec et al. [18] was based on Slovenian outpatient prescriptions in 2018 and used the ACB scale for the assessment of AB. Results were stratified by the age groups children (≤18 years), adults (19–64 years), and older adults (≥65 years) but not by sex within these groups. The overall prevalence of ACB≥3 in the Slovenian population was 7.6%, similar to our results (7.2% in men and 10.4% in women). Prevalence of use of at least one MAC in Slovenian children was 20.7% which was similar to our study (21.5% in boys and 20.9% in girls). However, prevalence of ACB≥3 was much lower in Slovenian children (1.2% vs. 5.6% in boys and 5.7% in girls). The prevalence of use of at least one MAC among adults in Slovenia was in the lower ranges of the German results (25.8% vs. 15.2%–47.5% in men and 22.6%–62.7% in women). However, the prevalence of ACB≥3 for adults was similar (7.3% vs. 1.7%–11.1% in men and 2.7%–14.8% in women). Interestingly, the prevalence of use of at least one MAC in Slovenian older adults was much lower than in Germany with 43.1% vs. 59.0%–71.1% in men and 62.7%–76.0% in women as was the prevalence of ACB≥3 with 12.1% vs. 17.2%–22.7% in men and 21.9%–26.3% in women. As the list of MACs used in our study is more extensive than the one used by Cebron Lipovec et al. it is not

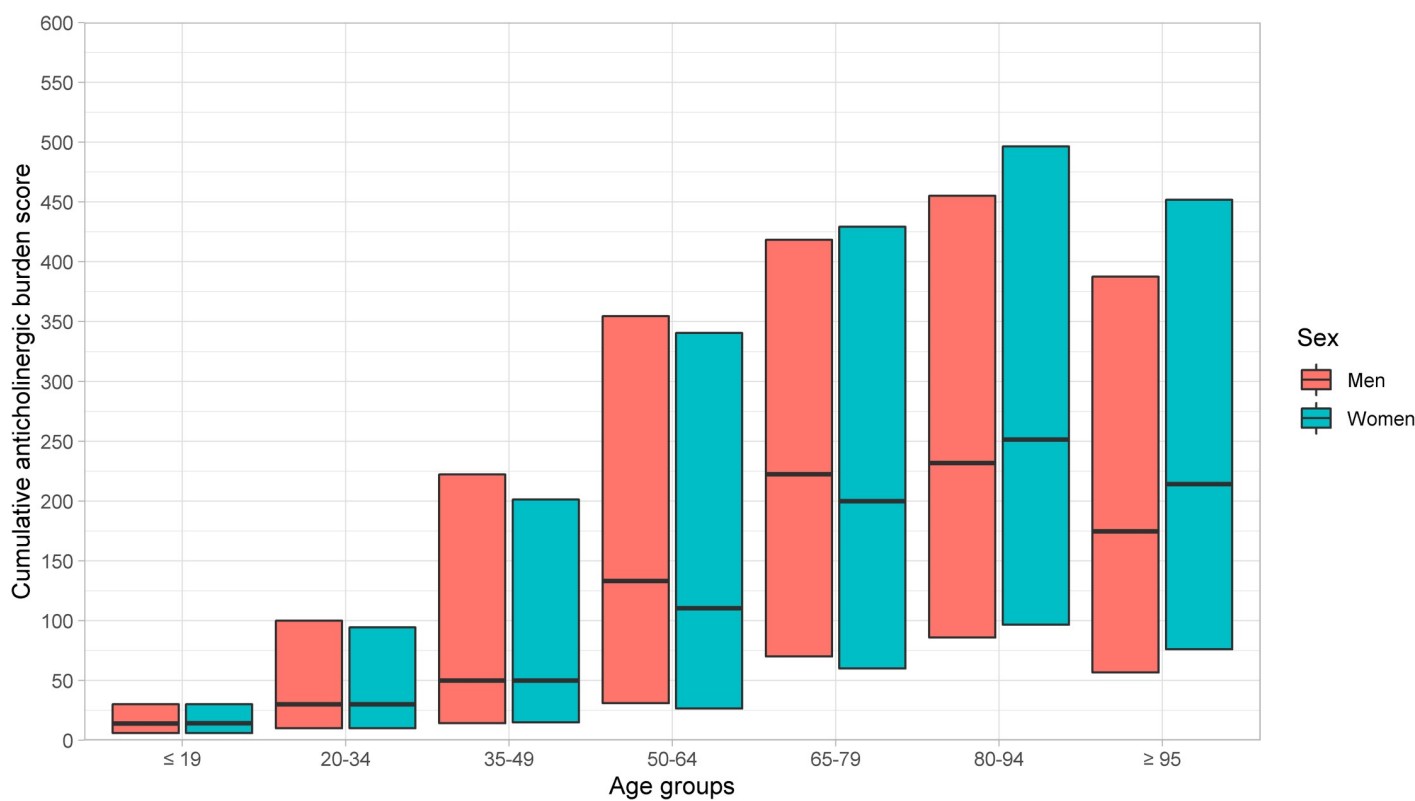

**Fig 4. Median (Q1-Q3) cumulative anticholinergic burden, by sex and age.**

clear whether the differences in prevalence of AB are due to the prescription behavior regarding MACs or the definition of MACs. However, the much lower prevalence of ACB≥3 in Slovenian older adults compared to German older adults is notable.

Among studies conducted in Germany, the comparison to our findings is hampered given that they were typically restricted to older adults or patients with a certain indication: Pfistermeister et al. [22] conducted a study in a population of hospitalized geriatric patients (median age 82 years), Ivchenko et al. [23] in older adults with overactive bladder (median age 75 years), Lippert et al. [24] in patients with dementia (mean age 84.7 years), Mayer et al. [25] in community-dwelling older German adults (median age 72 years), Phillips et al. [26] in community-dwelling older adults aged 65 years and older (mean age 73.8), and Mueller et al. [7] in patients undergoing cancer surgery (mean age 71.8 years). In the studies of Pfistermeister et al. [22] and Ivchenko et al. [23], where AB was assessed through ACB scale and categorized in the same way as in our study, the AB was similar, ACB≥3 27% and 25%, respectively, to the results of our study where the prevalence of an ACB≥3 was above 20% from age 70 in women and from age 80 in men. The studies of Lippert et al. [24] and Mayer et al. [25] also used the ACB scale but assessed AB as use of ≥1 MAC. The AB prevalence in their populations, 50% and 46%, respectively, was slightly lower than in our study (59%–71% in men, 63%–76% in women in the age groups 70 to ≥100 years). The studies of Phillips et al. [26] and Mueller et al. [7] reported much lower prevalences of AB, 19% and 16%, respectively, than our study. However, comparisons with the results of our study are difficult as Phillips et al. [26] used the Drug Burden Index (DBI) [27] and Mueller et al. [7] the Anticholinergic Drug Scale (ADS) [28] for the assessment of AB, which use different lists of MACs (e.g., unlike the ACB scale, the DBI does not consider inhaled MAC) and calculate AB differently (the DBI also includes the prescribed

**Table 3. Contribution of anticholinergic medication classes to cumulative anticholinergic burden[a] in men and women, with at least one dispensation of medication with anticholinergic activity (MAC) during the observation period (2016), by age group.**

| Characteristics | Age group[b] | | | | | | | | | | | | | |
| --- | --- | --- | --- | --- | --- | --- | --- | --- | --- | --- | --- | --- | --- | --- |
| | ≤ 19 | | 20–34 | | 35–49 | | 50–64 | | 65–79 | | 80–94 | | ≥ 95 | |
| | Men | Women | Men | Women | Men | Women | Men | Women | Men | Women | Men | Women | Men | Women |
| | N = 306,894 | N = 288,767 | N = 260,776 | N = 378,155 | N = 372,387 | N = 568,948 | N = 627,682 | N = 913,551 | N = 609,698 | N = 848,695 | N = 221,985 | N = 373,292 | N = 4,021 | N = 16,830 |
| **MAC class (%)[c]** | | | | | | | | | | | | | | |
| Antidepressants | 8.1% | 16.3% | 38.7% | 47.9% | 35.9% | 45.3% | 25.6% | 36.1% | 11.8% | 21.4% | 10.3% | 19.4% | 11.2% | 18.9% |
| Antihistamines | 24.0% | 20.7% | 3.6% | 4.4% | 2.0% | 2.8% | 1.2% | 1.7% | 0.9% | 1.2% | 1.3% | 1.7% | 2.7% | 2.4% |
| Antipsychotics | 3.9% | 3.1% | 18.1% | 8.3% | 14.6% | 9.1% | 6.4% | 6.2% | 2.1% | 2.8% | 1.6% | 2.4% | 2.4% | 3.5% |
| Benzodiazepines | 0.6% | 0.6% | 1.0% | 0.7% | 1.0% | 0.9% | 0.8% | 0.9% | 0.6% | 1.0% | 0.8% | 1.3% | 1.6% | 2.3% |
| Cardiovascular medication | 0.9% | 0.8% | 2.5% | 2.4% | 7.6% | 5.8% | 16.5% | 12.2% | 23.8% | 20.8% | 25.4% | 23.3% | 26.0% | 22.5% |
| Diuretics | 0.2% | 0.2% | 0.3% | 0.4% | 1.1% | 1.3% | 2.9% | 2.7% | 5.6% | 5.5% | 9.8% | 9.8% | 19.0% | 16.7% |
| Gastrointestinal medication | 0.7% | 0.9% | 1.0% | 1.2% | 1.5% | 1.3% | 1.9% | 1.7% | 2.1% | 2.0% | 1.9% | 1.8% | 1.8% | 1.6% |
| Opioids | 0.4% | 0.5% | 2.0% | 1.6% | 3.1% | 2.5% | 2.9% | 2.8% | 2.5% | 3.3% | 3.0% | 5.3% | 4.5% | 8.2% |
| Medication for Parkinson's disease | 0.2% | 0.1% | 0.2% | 0.3% | 0.5% | 0.5% | 1.4% | 1.0% | 4.0% | 2.5% | 5.3% | 2.9% | 2.1% | 1.7% |
| Medication for urinary incontinence/overactive bladder | 8.9% | 5.6% | 2.8% | 3.2% | 2.4% | 4.0% | 3.4% | 6.0% | 8.9% | 10.7% | 13.5% | 12.5% | 12.1% | 10.8% |
| Medication for respiratory diseases | 2.9% | 2.4% | 1.6% | 2.3% | 2.9% | 3.2% | 5.0% | 4.6% | 6.4% | 4.9% | 6.2% | 3.6% | 4.9% | 2.6% |
| Glucocorticoids | 12.5% | 12.3% | 9.4% | 10.7% | 7.5% | 8.5% | 6.4% | 7.1% | 6.5% | 6.8% | 6.3% | 5.4% | 5.1% | 3.6% |
| Tropane alkaloids | 2.0% | 1.9% | 0.1% | 0.1% | 0.1% | 0.0% | 0.1% | 0.0% | 0.1% | 0.0% | 0.1% | 0.0% | 0.1% | 0.0% |
| Immunosuppressants | 1.4% | 1.1% | 1.8% | 1.3% | 1.1% | 0.8% | 0.6% | 0.5% | 0.3% | 0.3% | 0.2% | 0.1% | 0.1% | 0.1% |
| Muscle relaxants | 0.8% | 0.7% | 1.4% | 1.3% | 1.4% | 1.5% | 1.0% | 1.1% | 0.5% | 0.5% | 0.3% | 0.3% | 0.1% | 0.2% |
| Antiemetics | 0.8% | 1.1% | 0.8% | 1.1% | 0.4% | 0.6% | 0.3% | 0.5% | 0.3% | 0.5% | 0.4% | 0.7% | 0.6% | 0.9% |
| Antibiotics | 20.1% | 21.4% | 4.6% | 5.7% | 2.1% | 2.2% | 1.1% | 1.3% | 1.1% | 1.2% | 1.1% | 0.9% | 1.1% | 0.9% |
| Antiepileptics | 9.9% | 7.1% | 7.0% | 3.3% | 5.0% | 2.7% | 2.9% | 2.0% | 1.5% | 1.1% | 1.0% | 0.7% | 0.8% | 0.3% |
| Non-opioid analgesics | 0.2% | 0.5% | 1.1% | 1.2% | 1.6% | 1.8% | 1.6% | 2.1% | 1.1% | 1.7% | 0.9% | 1.3% | 0.9% | 0.9% |
| Antidiabetics | 0.1% | 0.3% | 1.1% | 1.3% | 7.1% | 3.4% | 16.6% | 7.4% | 18.6% | 10.3% | 10.1% | 6.0% | 2.8% | 1.8% |

[a] The cumulative anticholinergic burden of a MAC is calculated by multiplying each MAC's anticholinergic cognitive burden (ACB) score by its duration (prescribed number of DDDs).

[b] The total anticholinergic burden per age group is calculated by summing up each person's cumulative anticholinergic burden during the observation period in the respective age group.

[c] Cumulative anticholinergic burden stratified by MAC class.

**Table 4. Contribution of prescriber specialty to cumulative anticholinergic burden[a] in men and women, with at least one dispensation of medication with anticholinergic activity (MAC) during the observation period (2016), by sex and age group.**

| Characteristics | Age group[b] | | | | | | | | | | | | | |
|---|---|---|---|---|---|---|---|---|---|---|---|---|---|---|
| | ≤19 | | 20–34 | | 35–49 | | 50–64 | | 65–79 | | 80–94 | | ≥95 | |
| | Men | Women | Men | Women | Men | Women | Men | Women | Men | Women | Men | Women | Men | Women |
| | N = 306,894 | N = 288,767 | N = 260,776 | N = 378,155 | N = 372,387 | N = 568,948 | N = 627,682 | N = 913,551 | N = 609,698 | N = 848,695 | N = 221,985 | N = 373,292 | N = 4,021 | N = 16,830 |
| **Prescribers of MAC (%)[c]** | | | | | | | | | | | | | | |
| General practitioner | 58.3% | 54.1% | 39.2% | 41.0% | 48.5% | 47.0% | 63.5% | 56.5% | 71.2% | 68.5% | 73.3% | 78.2% | 85.0% | 86.3% |
| Anesthesiology | 0.1% | 0.1% | 0.3% | 0.4% | 0.5% | 0.8% | 0.6% | 1.0% | 0.3% | 0.6% | 0.2% | 0.3% | 0.1% | 0.1% |
| Ophthalmology | 5.0% | 5.5% | 2.3% | 2.5% | 1.6% | 1.5% | 1.2% | 1.3% | 1.9% | 2.2% | 2.0% | 1.6% | 1.1% | 0.6% |
| Surgery | 0.8% | 0.7% | 1.0% | 1.0% | 1.2% | 1.4% | 1.0% | 1.3% | 0.6% | 0.9% | 0.4% | 0.5% | 0.2% | 0.2% |
| Gynecology | 0.0% | 0.3% | 0.0% | 1.1% | 0.0% | 1.0% | 0.0% | 1.3% | 0.0% | 1.8% | 0.0% | 0.9% | 0.0% | 0.2% |
| Otorhinolaryngology | 2.0% | 1.7% | 1.3% | 1.4% | 0.9% | 0.8% | 0.5% | 0.4% | 0.2% | 0.2% | 0.1% | 0.1% | 0.0% | 0.0% |
| Dermatology | 7.5% | 8.7% | 2.3% | 3.2% | 0.9% | 1.1% | 0.5% | 0.6% | 0.3% | 0.3% | 0.3% | 0.2% | 0.3% | 0.3% |
| Internal medicine | 1.6% | 1.7% | 4.2% | 4.5% | 4.5% | 4.6% | 6.1% | 5.8% | 7.3% | 6.1% | 5.7% | 3.5% | 2.5% | 1.5% |
| Pediatrics | 5.6% | 4.8% | 0.1% | 0.1% | 0.0% | 0.0% | 0.0% | 0.0% | 0.0% | 0.0% | 0.0% | 0.0% | 0.0% | 0.0% |
| Psychology and psychiatry | 5.9% | 9.7% | 27.7% | 26.7% | 25.3% | 26.0% | 15.9% | 19.9% | 6.8% | 9.0% | 5.2% | 6.0% | 2.9% | 5.3% |
| Neurosurgery | 0.0% | 0.0% | 0.1% | 0.1% | 0.2% | 0.2% | 0.1% | 0.1% | 0.1% | 0.1% | 0.0% | 0.0% | 0.0% | 0.0% |
| Neurology | 0.6% | 0.9% | 5.1% | 5.3% | 4.8% | 5.2% | 3.5% | 4.0% | 3.1% | 2.9% | 2.8% | 2.2% | 1.8% | 1.6% |
| Radiology | 0.0% | 0.0% | 0.0% | 0.0% | 0.0% | 0.1% | 0.0% | 0.1% | 0.0% | 0.0% | 0.0% | 0.0% | 0.1% | 0.0% |
| Physical medicine and rehabilitation | 0.0% | 0.0% | 0.1% | 0.1% | 0.1% | 0.1% | 0.1% | 0.1% | 0.0% | 0.1% | 0.0% | 0.1% | 0.0% | 0.0% |
| Urology | 3.3% | 2.1% | 1.6% | 2.0% | 1.5% | 2.2% | 2.4% | 3.0% | 6.4% | 5.1% | 8.7% | 4.7% | 5.1% | 2.1% |
| Unknown specialty | 9.8% | 10.1% | 15.6% | 11.4% | 11.1% | 9.1% | 5.6% | 5.7% | 2.4% | 2.7% | 1.6% | 1.9% | 1.2% | 2.0% |

[a] The cumulative anticholinergic burden of a MAC is calculated by multiplying each MAC's anticholinergic cognitive burden (ACB) score by its duration (prescribed number of DDDs).

[b] The total anticholinergic burden per age group is calculated by summing up each person's cumulative anticholinergic burden during the observation period in the respective age group.

[c] Cumulative anticholinergic burden stratified by prescribing physician's specialty.

dose). Furthermore, in the study of Phillips et al. [26], there might have been a selection of healthier patients into the study population as suggested by their non-responder analysis.

Our study provides information on the use of MAC and AB across all age groups. This analysis showed that use of MAC in Germany can roughly be divided into four phases: (i) persons aged ≤19 years with a low cumulative AB mainly due to use of antihistamines, antibiotics, and glucocorticoids; (ii) persons aged 20–49 years with a low but steadily increasing cumulative AB with antidepressants as the main contributor to the cumulative AB; (iii) a transitional phase in persons aged 50–64 where the contribution of cardiovascular medication and antidiabetics starts to increase, which is higher in men than in women; and (iv) persons aged ≥65 years where the relative contribution of antidepressants decreases due to the increased contribution of medication for the treatment of cardiovascular disease, diabetes, and urinary incontinence/overactive bladder. The increased burden of chronic diseases is reflected in the high cumulative AB, which peaks in the age group 80–94 years.

MAC prescribed by general practitioners accounted for 39–86% of the total cumulative AB and thus had the highest share. In health systems with the general physician in the role of gatekeeper, this proportion might be even higher. In Germany, persons are free to choose which physician to see. There is no requirement of a referral from a general practitioner to access specialist care. In our study, there was an age gradient regarding the diversity of physician specialties contributing to the cumulative AB. In the oldest age groups, MACs were almost exclusively prescribed by general practitioners. In Germany, patients in these age groups are also treated by specialists but refills of medication are often prescribed by general practitioners. Therefore, this result is to be expected. These aspects are relevant if interventions to reduce the AB in specific patient groups or to increase the awareness of AB in general were to be designed. Our results suggest that general practitioners would be an important target group, particularly for older age groups but involvement of specialists, who often initiate prescriptions of a certain medication, may also be required.

Our study showed that there are persons with an AB considered to be clinically relevant in all age groups. This demonstrates the need to conduct studies on potentially harmful effects not only in older adults but also in children, adolescents, and the entire adult population. However, it has to be kept in mind that there are a lot of unanswered questions in regards to how AB can cause or contribute to clinically relevant adverse effects. For example, the time period over which the cumulative effects of anticholinergic burden may accrue and possibly produce harms are unclear. Also the role of type and dosage of single MACs and their overlap are not well understood. When planning a study on the risk of AB, this means that classifying persons as exposed or unexposed bears a high level of uncertainty, so robustness of findings would need to be assessed by comprehensive sensitivity analyses. Also in many other regards, studies on the risk of outcomes associated with AB are challenging, e.g. regarding issues such as confounding by indication, unmeasured confounding and time-varying exposure.

To our knowledge this is the first study in Germany providing a detailed description of the AB in an unselected population sample, i.e., without restrictions to a certain age or patient group. The large sample size allowed us to precisely estimate the prevalence of the AB stratified by age and sex. AB was estimated using the ACB scale—a widely used and validated tool—and a list of MACs created specifically for the German health care system. There are many scales for the assessment of AB and they have been shown to differ [29, 30]. Thus, direct comparisons with studies using other AB scales are difficult. Moreover, medications classified with an ACB score of 1 only have a possible anticholinergic effect based on in vitro affinity to muscarinic receptors without clinically relevant negative cognitive effects. It is not clear whether the cumulative use of several medications with a possible anticholinergic effect is equivalent to the AB induced through the use of medications with established and clinically relevant cognitive

anticholinergic effects (ACB scores 2 or 3). However, some studies have shown increased risks of adverse effects already for an ACB score of 1 [22, 31].

Our study was based on German claims data. Due to the nature of the data the study is not affected by recall or volunteer bias. Moreover, the study population was fairly stable: 91% of included persons were observable for the whole year of 2016, 98% were observable for 90 days or more and only 3.3% exited the study before the end of the observation period due to end of continuous insurance. Limitations of the data source include lack of information regarding the use of medication during hospitalization as well as lack of information on adherence—no information is available on whether dispensed medication was actually used by the patient. Furthermore, over-the-counter medication is not captured, thus dispensations of MACs, particularly of antihistamines, might have been underestimated. Treatment durations of MACs were estimated using DDDs as the prescribed dose is not available. However, for each MAC we reviewed summaries of product characteristics and, if applicable, adapted lower DDDs for persons aged <18 and ≥65 years. Nonetheless, this approach is not equivalent to other studies that had more information on dosage and used more sophisticated methods to take it into account. Finally, in our study we have not assessed AB in a longitudinal manner, which–in view of the aforementioned unanswered questions about clinically relevant AB levels–would be essential in a subsequent risk study to understand the potential link between AB exposure and negative health outcomes. Such risk studies are particularly needed in the younger population where it is even less clear if such a link exists at all.

In conclusion, this comprehensive overview showed that a clinically relevant AB is common in the German general population. This holds particularly true for older persons but there are also younger age groups with a prevalence of up to 7%. Among adults, prevalence of clinically relevant AB was consistently higher in women than in men. Given the known risks associated with AB in older persons, targeted interventions at the prescriber level are needed. Furthermore, studies exploring possible risks associated with AB in children, adolescents and the entire adult population are warranted.

## Supporting information

**S1 Table. Description of study population stratified by anticholinergic burden measured through Anticholinergic Cognitive Burden (ACB) score.**
(DOCX)

## Author Contributions

**Conceptualization:** Jonas Reinold, Oliver Riedel, Ulrike Haug.

**Data curation:** Malte Braitmaier.

**Formal analysis:** Malte Braitmaier.

**Methodology:** Jonas Reinold, Oliver Riedel, Ulrike Haug.

**Project administration:** Jonas Reinold, Malte Braitmaier, Oliver Riedel, Ulrike Haug.

**Resources:** Ulrike Haug.

**Software:** Jonas Reinold, Malte Braitmaier.

**Supervision:** Oliver Riedel, Ulrike Haug.

**Validation:** Jonas Reinold, Malte Braitmaier.

**Visualization:** Jonas Reinold.

**Writing – original draft:** Jonas Reinold.

**Writing – review & editing:** Jonas Reinold, Malte Braitmaier, Oliver Riedel, Ulrike Haug.

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
