## [Decision Letter · Decision Letter 0]

12 Feb 2021

PONE-D-20-37319

Anticholinergic burden: first comprehensive analysis using claims data shows large variation by age and sex

PLOS ONE

Dear Dr. Reinold,

Thank you for submitting your manuscript to PLOS ONE. After careful consideration, we feel that it has merit but does not fully meet PLOS ONE’s publication criteria as it currently stands. Therefore, we invite you to submit a revised version of the manuscript that addresses the points raised during the review process.

I would like to thank you for your patience while your manuscript was under review. As I'm sure you are aware, the on-going pandemic is impacting everyone's availability for activities like manuscript reviews. We now have 2 reviews in of your manuscript. You will see that the reviewers have identified the potential contribution of this research, in particular the inclusion of younger populations in the assessment of anticholinergic burden. Both reviewers, also raise some addition and important points for you to consider in both the analysis and the presentation of your work (please see what Reviewer #1 suggests). In particular, you will note that Reviewer #2 has raised some points about incorporating dose and co-use in the assessment of anticholinergic burden. If you chose to resubmit, I would suggest that you pay careful attention to these points in revisions to the analysis, your response, or both.

We look forward to receiving your revised manuscript.

Kind regards,

Andrea Gruneir

Academic Editor

PLOS ONE

Journal Requirements:

"UH, OR, MB and JR are working at an independent, non-profit research institute, the Leibniz Institute for Prevention Research and Epidemiology – BIPS. Unrelated to this study, BIPS occasionally conducts studies financed by the pharmaceutical industry. Almost exclusively, these are post-authorization safety studies (PASS) requested by health authorities. The design and conduct of these studies as well as the interpretation and publication are not influenced by the pharmaceutical industry. The study presented was not funded by the pharmaceutical industry. The authors have no relevant financial or non-financial interests to disclose."

Reviewers' comments:

Reviewer's Responses to Questions

**Comments to the Author**

1. Is the manuscript technically sound, and do the data support the conclusions?

Reviewer #1: Yes

Reviewer #2: Partly

2. Has the statistical analysis been performed appropriately and rigorously? 

Reviewer #1: Yes

Reviewer #2: Yes

3. Have the authors made all data underlying the findings in their manuscript fully available?

Reviewer #1: No

Reviewer #2: Yes

4. Is the manuscript presented in an intelligible fashion and written in standard English?

Reviewer #1: Yes

Reviewer #2: Yes

5. Review Comments to the Author

Reviewer #1: The authors present a descriptive cohort study to describe the anticholinergic burden among a general population from German. The authors found identified that anticholinergic burden (ACB) increased in females and older patients, which is well known from other studies. However, the main contribution to the literature are the findings in younger populations, which are often excluded in studies of ACB. Additionally, the analyses stratifying by medication class and prescriber are interesting, and can be important to help identify target groups for interventions to minimize ACB in practice.

Major comments

1. In the write-up of the study design and study population, it would be helpful if the authors specified the study design used as per the STROBE guidelines.

2. It may also be helpful if the authors were to provide a graphical depiction of the study design, particularly relating to the inclusion and exclusion criteria (For examples, see Schneeweiss S. et al. Ann Int Med 2019. PMID: 30856654)

3. A study flow diagram of patient inclusion would be useful (as per STROBE guidelines).

4. There seems to be a disconnect between the description in text and choice for supplementary vs. in-text tables. For example, the information presented in the appendix table contributes to a substantial portion of the text, and even one of the main conclusions in the first paragraph of the discussion. Conversely, very little attention is paid to Table 1 in the manuscript. I would consider to switch these tables, or consider combining both in one larger table?

5. Similar to the above, perhaps the authors clarify the purpose of Table 1?

6. How the medications were identified in Table 2? Here, I believe these are only the medications that were dispensed in the observation period in 2016? This does or does not include prescriptions dispensed prior to baseline? Perhaps a study design diagram could help clarify this, as recommended above.

7. The proportions presented in Table 3 are not clear to me. I believe this is described on page 3 lines 71-74 in the methods section. However, it would be useful if the authors provided an example of the calculation (or the written algorithm). Additionally, is this a validated approach or one developed by the authors?

Specific Comments

(Note the line numbers changed throughout the document, making it challenging to identify exact location of comments)

1. Page 2 Line 43 (Methods): What do OPS and EBM stand for? I could not find these acronyms defined previously.

2. Page 1 Line 1 (Results): For consistency with the following sentence, I would recommend changing “prevalences decreased with decreasing ACB score” to “prevalence increased with increasing ACB score”. Please also specify that these are prior morbidities and medications.

3. Page 1 line 10 (Discussion): Previous the acronym “AB” was used for “anticholinergic burden”, however, here it is spelled out. This should likely be switched to “AB”.

4. Page 1 line 19 (discussion): I believe there should be a comma after “interestingly”.

Reviewer #2: Thank you for the opportunity to review this interesting manuscript; which describes the assessment of anticholinergic burden among a large sample of Germans. The data source and sample are well described.

My main concern is with the measure of exposure - which is essentially supposed to be cumulative anticholinergic burden.

o Why limit to just one year of follow-up? There is a risk of underascertainment of exposure, particularly for anticholinergic medications dispensed just prior to the start of 2016.

o As I read the manuscript, the minimum follow up was only one day; this also could contribute to bias in the measure of exposure. Provide some justification for this; also perform sensitivity analyses on limiting to individual with a reasonable amount of follow up.

o When assessing cumulative anticholinergic exposure it is important to understand all of which medications are used, the dosage at which they are taken, and the patterns/overlap in which different medications are taken  and all of these, over the long term. Aside from the already-identified issues with minimum and maximum follow up time for the cohort, dosage is inadequately addressed using the current methodology. A variety of publications exist that describe methods to estimate cumulative anticholinergic burden more precisely and these could be employed here.

After reviewing the methods and results, I am left with the feeling that there is room for inaccuracy -- and likely, underestimation -- in cumulative anticholinergic assessment. I think the authors could benefit from considering those other methods and also using sensitivity analyses to explore the impact on results.

With respect to outcomes, the prevalence of anticholinergic burden is interesting -- but really only as an intermediate on the pathway to the negative health outcomes that are the end result of high cumulative anticholinergic burden. The time period of the study is insufficient to truly understand the risk of these long term effects; and why such a short time period was selected was not expanded upon in the manuscript. Nonetheless, if the rationale for this study is that anticholinergic burden in young people is under-studied, demonstrating that this is important (because it is associated with negative health outcomes in young people) would be an important step. I think that whether anticholinergic exposure is an issue among young people remains to be established and this manuscript could delve more in to that

Minor points

-The manuscript could benefit from some proofreading/English language editing.

-The discussion is wordy and could be streamlined; particularly the paragraphs that repeat the results in comparing to the published literature.

6. PLOS authors have the option to publish the peer review history of their article (what does this mean?). If published, this will include your full peer review and any attached files.

Reviewer #1: No

Reviewer #2: No

---

## [Author Response · Author response to Decision Letter 0]

25 Mar 2021

We look forward to receiving your revised manuscript.

Kind regards,

Andrea Gruneir

Academic Editor

PLOS ONE

Journal Requirements:

We thank the editor for this remark. We changed the funding information from “This study was funded entirely by internal funds of the Leibniz Institute for Prevention Research and Epidemiology – BIPS. The funding institution had no influence on any part of this article.” to “The author(s) received no external funding for this work.”

"UH, OR, MB and JR are working at an independent, non-profit research institute, the Leibniz Institute for Prevention Research and Epidemiology – BIPS. Unrelated to this study, BIPS occasionally conducts studies financed by the pharmaceutical industry. Almost exclusively, these are post-authorization safety studies (PASS) requested by health authorities. The design and conduct of these studies as well as the interpretation and publication are not influenced by the pharmaceutical industry. The study presented was not funded by the pharmaceutical industry. The authors have no relevant financial or non-financial interests to disclose."

 We thank the editor for this suggestion. We have added the suggested phrase to the COI statement “Moreover, this does not alter our adherence to PLOS ONE policies on sharing data and materials.”

 We thank the editor for this suggestion. We modified the section “Availability of data and material” as follows and included it in the updated version of the cover letter.

In Germany, use of personal data is protected by the Federal Data Protection Act and particularly the use of claims data for research is regulated by the Code of Social Law. Researchers have to apply for a project-specific permit from the statutory health insurance providers which then need an approval from their governing authorities. The use of the data on which this publication is based was only allowed for BIPS employees within the framework of the specified project and limited to a pre-defined time span. Researchers who want to access the data on which this publication is based need to ask for new approval by the statutory health insurance providers DAK-Gesundheit (service@dak.de), die Techniker (service@tk.de), hkk Krankenkasse (info@hkk.de) and AOK Bremen/Bremerhaven (info@hb.aok.de) which upon granting approval would have to ask their respective authorities for approval. Please contact gepard@leibniz-bips.de for help with this process. The authors confirm that they had no special access privileges to the data and that other researchers will be able to access the data in the same manner as the authors by following the instructions described above.

 We thank the editor for pointing this out. The paragraph in question has been rewritten and the phrase “data not shown” is no longer in the manuscript.

 We thank the editor for this remark. We have moved the ethics statement to the methods section (Page 4, Line 109-115)

Reviewers' comments:

Reviewer's Responses to Questions

Comments to the Author

1. Is the manuscript technically sound, and do the data support the conclusions?

Reviewer #1: Yes

Reviewer #2: Partly

2. Has the statistical analysis been performed appropriately and rigorously? 

Reviewer #1: Yes

Reviewer #2: Yes

3. Have the authors made all data underlying the findings in their manuscript fully available?

Reviewer #1: No

Reviewer #2: Yes

4. Is the manuscript presented in an intelligible fashion and written in standard English?

Reviewer #1: Yes

Reviewer #2: Yes

5. Review Comments to the Author (Please note that all references to pages and lines refer to the clean version of the manuscript)

Reviewer #1: The authors present a descriptive cohort study to describe the anticholinergic burden among a general population from German. The authors found identified that anticholinergic burden (ACB) increased in females and older patients, which is well known from other studies. However, the main contribution to the literature are the findings in younger populations, which are often excluded in studies of ACB. Additionally, the analyses stratifying by medication class and prescriber are interesting, and can be important to help identify target groups for interventions to minimize ACB in practice.

Major comments

1. In the write-up of the study design and study population, it would be helpful if the authors specified the study design used as per the STROBE guidelines.

 We thank the reviewer for this remark. We now specify the study design in the section “study design and study population” (page 2, line 55): 

 Page 2, line 55: We conducted a cross-sectional study using data from the year 2016, the most recent data at the time of analysis.

2. It may also be helpful if the authors were to provide a graphical depiction of the study design, particularly relating to the inclusion and exclusion criteria (For examples, see Schneeweiss S. et al. Ann Int Med 2019. PMID: 30856654)

 We thank the reviewer for this suggestion. We have created a graphical depiction of the study design (Figure 1). The example by Schneeweiss et al. is very clear and nicely illustrates the inclusion and exclusion criteria. As the example by Schneeweis et al. refers to a longitudinal study design whereas our study has a cross-sectional design, we adapted the template to make sure that there is no misunderstanding regarding the study design, while making sure that the inclusion and exclusion criteria are clearly depicted. 

3. A study flow diagram of patient inclusion would be useful (as per STROBE guidelines).

 We have followed the reviewer’s advice and have created a flow diagram to visualize the inclusion and exclusion process (Figure 2).

4. There seems to be a disconnect between the description in text and choice for supplementary vs. in-text tables. For example, the information presented in the appendix table contributes to a substantial portion of the text, and even one of the main conclusions in the first paragraph of the discussion. Conversely, very little attention is paid to Table 1 in the manuscript. I would consider to switch these tables, or consider combining both in one larger table?

5. Similar to the above, perhaps the authors clarify the purpose of Table 1?

 We agree with the reviewer’s remarks. We have switched tables T1 and S1. The purpose of S1 (formerly T1) was to generally describe patients regarding health care utilization, morbidities and medication stratified by anticholinergic burden. As expected, patients with higher anticholinergic burden more often had chronic disease and more often received various types of pharmacological therapy.

6. How the medications were identified in Table 2? Here, I believe these are only the medications that were dispensed in the observation period in 2016? This does or does not include prescriptions dispensed prior to baseline? Perhaps a study design diagram could help clarify this, as recommended above.

 We thank the reviewer for this comment. Table 2 does indeed contain only medications that were dispensed during the study period in 2016. In order to make this clearer we have followed the reviewer’s advice and created a graphical depiction of the study design (new Figure 1) and now also stress it in the methods section. Moreover, we have modified the title of Table 2 to “Prevalence of use of medications with anticholinergic activity (MACs) in persons with anticholinergic burden measured through the anticholinergic cognitive burden (ACB) scale during the observation period (2016)”

7. The proportions presented in Table 3 are not clear to me. I believe this is described on page 3 lines 71-74 in the methods section. However, it would be useful if the authors provided an example of the calculation (or the written algorithm). Additionally, is this a validated approach or one developed by the authors?

 We thank the reviewer for this suggestion. We fully agree that the explanation how the cumulative AB score was calculated becomes clearer if it is illustrated by an example. Therefore, we have now inserted an example in the methods section. Furthermore, we now mention that our calculations follow the approach proposed by Campbell et al. and we explain that Campbell et al. additionally calculated a mean score per person whereas we focused on the total score and did not further transfer it into a mean (page 3, lines 91-99)

 Page 3, lines 91-99: In order to describe the cumulative AB of each person during the observation period, we first multiplied the AB score of each MAC dispensed to the person during the observation period or overlapping the observation period with the length of supply (based on DDD) and then summed up the score points of all dispensations. Subsequently, these AB scores were summed up per person to calculate the cumulative AB. For example, a person receiving 200 DDDs of metformin (ACB score 1) and 30 DDDs of tramadol (ACB score 2) during the observation period had a cumulative AB of 260 (i.e., the result of 200 x 1 + 30 x 2). This method was proposed by Campbell et al. [1]. Campbell et al. further divided the cumulative AB by the number of days in the exposure period to transfer the total AB score into a mean score per person but this additional transformation was not relevant in the context of our study [1]..

Specific Comments

(Note the line numbers changed throughout the document, making it challenging to identify exact location of comments)

 We thank the reviewer for pointing out this mistake. The line numbers were changed to a continuous enumeration

1. Page 2 Line 43 (Methods): What do OPS and EBM stand for? I could not find these acronyms defined previously.

 We thank the reviewer for this remark. OPS stands for “Operationen- und Prozedurenschlüssel” which roughly translates to “Operations and procedure classification” and EBM-“Einheitlicher Bewertungsmaßstab” translates to “Uniform Value Scale”. We have chosen to remove the acronyms OPS and EBM from the text, as they do not provide any additional information to international readers (page 2, lines 64-66).

 Page 2, lines 64-66: Morbidities were assessed any time prior to observation period (starting from 2004) through records of ≥1 ICD-10-GM inpatient or outpatient diagnoses or records of ≥1 codes of relevant operations, procedures or outpatient services as well as participation in disease management plans.

2. Page 1 Line 1 (Results): For consistency with the following sentence, I would recommend changing “prevalences decreased with decreasing ACB score” to “prevalence increased with increasing ACB score”. Please also specify that these are prior morbidities and medications.

 We thank the reviewer for this suggestion. We have followed it and changed the sentence accordingly (page 7 lines 134-135).

Page 7 lines 134-135: For all morbidities and medications assessed prior to start of observation period, prevalences increased with increasing ACB score (S1 Table).

3. Page 1 line 10 (Discussion): Previous the acronym “AB” was used for “anticholinergic burden”, however, here it is spelled out. This should likely be switched to “AB”.

 We thank the reviewer for pointing this out. We have abbreviated Anticholinergic burden to AB in the discussion

4. Page 1 line 19 (discussion): I believe there should be a comma after “interestingly”.

 We thank the reviewer for spotting this mistake, we have corrected it accordingly.

Reviewer #2: 

Thank you for the opportunity to review this interesting manuscript; which describes the assessment of anticholinergic burden among a large sample of Germans. The data source and sample are well described.

My main concern is with the measure of exposure - which is essentially supposed to be cumulative anticholinergic burden.

o Why limit to just one year of follow-up? There is a risk of underascertainment of exposure, particularly for anticholinergic medications dispensed just prior to the start of 2016.

 We apologize for the fact that we have obviously created a misunderstanding regarding our study design. We think the word follow-up was misleading and we now specify in the methods section that this was a cross-sectional study aimed to describe the prevalence of anticholinergic burden in Germany stratified by age and sex group in 2016. In addition, we have now added a graphical depiction of the study design to avoid this misunderstanding (new Figure 1). We have taken different approaches to describe the prevalence of AB because we think this provides a more comprehensive picture of the situation. On the one hand, we described the persons regarding the highest category of AB reached during the study period (i.e. in 2016). In addition, we calculated a measure that allowed us to assess the proportion of AB attributable to a certain class of MAC (e.g., antidepressants) or physician specialty. We called this measure “cumulative burden” because it takes into account all dispensations in the study period (i.e. in 2016). 

Regarding medication that was dispensed before the start of the study period, we included all anticholinergic medication that was dispensed in 2015 and whose treatment durations overlapped into the study period into calculation of the anticholinergic burden. In order to emphasize this point we modified the paragraph describing the assessment of exposure (pages 2-3, Lines 70-73): 

 Page 2-3, Line 70-73: Exposure to MAC was assessed based on outpatient prescriptions dispensed during the observation period, i.e., in 2016. Treatment durations were estimated based on DDDs. In case MAC were dispensed before 1 January 2016 and the days of supply covered by this dispensation overlapped with the observation period, the DDDs overlapping with the observation period were also considered.

o As I read the manuscript, the minimum follow up was only one day; this also could contribute to bias in the measure of exposure. Provide some justification for this; also perform sensitivity analyses on limiting to individual with a reasonable amount of follow up.

 We hope our explanation above clarifies that the word follow-up was misleading here as we conducted a cross-sectional study. Our intention was to avoid inclusion criteria that resulted in a non-inclusive selection of the study population as we wanted to provide a comprehensive overview of AB in an unselected population sample. We fear that using a continuous observation period in 2016 as inclusion criterion would have led to excluding certain subgroups, e.g. those who died in 2016. It is important to note that there is little fluctuation in GePaRD. In our study 91% of included persons were observable for the whole year 2016. 98% were observable for 90 days or more. Moreover, only 3.3% of persons exited the study before the end of the study period due to end of a continuous insurance period. Therefore, we do not see the duration of follow-up as a source of bias requiring additional analyses. We now mention this important aspect in the discussion (page 15, lines 248-250). 

 Page 15, lines 248-250: Moreover, the study population was fairly stable: 91% of included persons were observable for the whole year of 2016, 98% were observable for 90 days or more and only 3.3% exited the study before the end of the observation period due to end of continuous insurance.

o When assessing cumulative anticholinergic exposure it is important to understand all of which medications are used, the dosage at which they are taken, and the patterns/overlap in which different medications are taken  and all of these, over the long term. Aside from the already-identified issues with minimum and maximum follow up time for the cohort, dosage is inadequately addressed using the current methodology. A variety of publications exist that describe methods to estimate cumulative anticholinergic burden more precisely and these could be employed here.

 As explained above we hope it is now clearer why we calculated the “cumulative anticholinergic burden” here. It was just used as a measure to describe some further aspects of the prevalence of AB. We fully agree that for a study aiming to determine risk associated with a high AB, a more precise approach would be needed to determine exposed and unexposed person-time, but this was not the aim of this study. We now address this point in the discussion (page 16, lines 256-259). 

 Page 16, lines 256-259: Finally, cumulative AB was used in this study as a measure to further describe certain aspects regarding the prevalence of MAC prescribing. It should be kept in mind, however, that this measure might not be suitable for studies assessing the risk of outcomes associated with AB where a precise classification of exposed and unexposed time windows is important.

After reviewing the methods and results, I am left with the feeling that there is room for inaccuracy -- and likely, underestimation -- in cumulative anticholinergic assessment. I think the authors could benefit from considering those other methods and also using sensitivity analyses to explore the impact on results.

With respect to outcomes, the prevalence of anticholinergic burden is interesting -- but really only as an intermediate on the pathway to the negative health outcomes that are the end result of high cumulative anticholinergic burden. The time period of the study is insufficient to truly understand the risk of these long term effects; and why such a short time period was selected was not expanded upon in the manuscript. Nonetheless, if the rationale for this study is that anticholinergic burden in young people is under-studied, demonstrating that this is important (because it is associated with negative health outcomes in young people) would be an important step. I think that whether anticholinergic exposure is an issue among young people remains to be established and this manuscript could delve more in to that

 Thank you for this comment. As the reviewer states, the aim of this study was indeed to provide a comprehensive overview of the prevalence of AB by age and sex in an unselected population sample, including younger age groups which were understudied so far. We fully agree that risk studies would be needed in a next step to find out whether anticholinergic exposure is an issue among young people as our study only determined the prevalence of AB in this population. We now expand on this on page 14, Line 222-223

 Page 14, Line 222-223: This demonstrates the need to conduct studies on potentially harmful effects not only in older adults but also in children, adolescents, and the entire adult population.

Minor points

-The manuscript could benefit from some proofreading/English language editing.

 We thank the reviewer for this remark. The updated manuscript was proofread by a native speaker

-The discussion is wordy and could be streamlined; particularly the paragraphs that repeat the results in comparing to the published literature.

 We thank the reviewer for this suggestion. We have shortened the discussion section accordingly.

6. PLOS authors have the option to publish the peer review history of their article (what does this mean?). If published, this will include your full peer review and any attached files.

Do you want your identity to be public for this peer review? For information about this choice, including consent withdrawal, please see our Privacy Policy.

Reviewer #1: No

Reviewer #2: No

References

1. Campbell NL, Perkins AJ, Bradt P, Perk S, Wielage RC, Boustani MA, et al. Association of Anticholinergic Burden with Cognitive Impairment and Health Care Utilization Among a Diverse Ambulatory Older Adult Population. Pharmacotherapy: The Journal of Human Pharmacology and Drug Therapy. 2016;36(11):1123-31. doi: https://doi.org/10.1002/phar.1843.

---

## [Decision Letter · Decision Letter 1]

5 May 2021

PONE-D-20-37319R1

Anticholinergic burden: first comprehensive analysis using claims data shows large variation by age and sex

PLOS ONE

Dear Dr. Reinold,

Thank you for submitting your manuscript to PLOS ONE. After careful consideration, we feel that it has merit but does not fully meet PLOS ONE’s publication criteria as it currently stands. Therefore, we invite you to submit a revised version of the manuscript that addresses the points raised during the review process.

ACADEMIC EDITOR:

We now have 2 reviews back on your resubmitted manuscript. You will see that both reviewers stated that you have adequately addressed prior reviews but each have some remaining questions that they would like to see addressed (and I would agree). In particular, you will see that Reviewer #1 raises the issue of clarifying the study design including precise information on the timing of your measures; while Reviewer #2 asks for a more fulsome discussion of the cumulative measure and its use in younger adults. Reviewer #2 does acknowledge that many of their questions may not be answerable, it is clear that they are looking to see if a more thoughtful explanation of the utility of a cumulative measure (both the pros and the cons) as described in this study.

We look forward to receiving your revised manuscript.

Kind regards,

Andrea Gruneir

Academic Editor

PLOS ONE

Journal Requirements:

Reviewers' comments:

Reviewer's Responses to Questions

**Comments to the Author**

1. If the authors have adequately addressed your comments raised in a previous round of review and you feel that this manuscript is now acceptable for publication, you may indicate that here to bypass the “Comments to the Author” section, enter your conflict of interest statement in the “Confidential to Editor” section, and submit your "Accept" recommendation.

Reviewer #1: All comments have been addressed

Reviewer #2: (No Response)

2. Is the manuscript technically sound, and do the data support the conclusions?

Reviewer #1: Yes

Reviewer #2: Yes

3. Has the statistical analysis been performed appropriately and rigorously? 

Reviewer #1: Yes

Reviewer #2: Yes

4. Have the authors made all data underlying the findings in their manuscript fully available?

Reviewer #1: Yes

Reviewer #2: Yes

5. Is the manuscript presented in an intelligible fashion and written in standard English?

Reviewer #1: Yes

Reviewer #2: Yes

6. Review Comments to the Author

Reviewer #1: Thank you for addressing the majority of the comments raised during the initial review. I believe the authors have substantially improved the manuscript. However, I do have a couple of additional questions that I would appreciate if the authors could address further regarding the study design.

1. I do appreciate that the authors have tried to generate a figure for the study design, however, it is not quite clarifying the cross sectional design to me. Particularly regarding the follow-up and dates that the morbidities and medications were identified on. For example, if the inclusion was met on February 1st 2016 for a patient, I assume the morbidities were assessed from January 1st 2004 through to January 31st 2016 (one day before the inclusion date), and medications were identified from January 1st 2015 through to January 31st 2016 (again one day before inclusion)? This is the level of detail in the figure that would be helpful to better understand the design.

2. On the point of morbidity and medication assessment prior to inclusion, this information (e.g., what were to morbidities and medications) is not summarized in the paper? I imagine this information could be helpful to understand differences in the patients with AB. Therefore, I wonder why this is excluded from the results since it appears to be assessed?

2. I am still unsure if this is truly a cross-sectional design as it does not appear to be a single snapshot of a patient at a given time-point. But instead there appears to be follow-up. For example, in the methods section and Figure 1, it states that patients were "followed until (i) death, (ii) start of hospitalisation in 2016 with a duration of >=90-days, (iii) end of continuous insurance period, or (iv) end of observation period, whichever occurred first". Based on this statement, the design still reads like a cohort study as you are following individual patients over time. With this in mind I wonder if it is a descriptive cohort study rather than a true cross-sectional design? Can the authors further clarify this. I think it would also help address the concerns from both reviewers regarding the potential for misclassification on the AB.

I understand the authors aim of the paper is only to assess the prevalence of AB in a younger population, which is of high interest. And while a cohort study that compares outcomes is not the aim, I do think further clarification on the follow-up and time of measurements is still warranted so that future work can replicate these findings and potentially conduct a cohort study to assess health outcomes.

Reviewer #2: Thank you for the revisions. I think the study design is much clearer now and the explanatory figures included helpful.

The work done to explain that its really a cross-sectional prevalence (assessed over a one year period) is clarifying; but at the same time, I feel the implications of this requires some discussion or at least some further expansion in the limitations.

To the best of my knowledge (but this would be worth the authors discussing in any case) the time period over which the cumulative effects of anticholinergic burden accrue are unclear. The important point about high anticholinergic burden -- as the authors are well aware -- is not that people are on a number of medications, but rather than these medications together create an exposure that results in negative health outcomes like falls and dementia. But over what time period does this exposure need to occur? Does it matter if someone has a high level of anticholinergic burden for a 30 day period while a set of medications interact? Or does that cumulative exposure need to be over a 30 day period or 60 day period or for >1 year? How does the study design chosen illuminate the relationship between exposure and outcomes; or in another way, do we know that the exposures measured and reported here are clinically relevant?

To be clear, I don't think the authors need to actually answer all the questions above (and indeed, I believe some are presently unanswerable); but rather this is the context in which to discuss the strengths and limitations of their study design. I am unclear why reporting the results of a cross sectional measure of 'cumulative' (as the authors define is) is an important finding -- but i would welcome the authors defense of why this is a valid and important measure, through expansion of the discussion section.

At the very least, I would think a clear limitation to the design should be added to state that longitudinal assessment of anticholinergic burden that accounts for dose is what would be needed to understand the link between anticholinergic burden exposure and negative health outcomes -- particularly in the younger population who may have higher than expected anticholinergic burden but to my knowledge have never been shown to develop the negative health outcomes observed in older adults.

I recognize that the authors have inserted this limitation related to the above: 'It should be kept in mind, however, that this measure might not be suitable for studies assessing the risk of outcomes associated with AB where a precise classification of exposed and unexposed time windows is important.' However in my mind, that is the important goal of this type of work. So discussing/addressing that limitation more fulsomely would be useful.

7. PLOS authors have the option to publish the peer review history of their article (what does this mean?). If published, this will include your full peer review and any attached files.

Reviewer #1: No

Reviewer #2: No

---

## [Author Response · Author response to Decision Letter 1]

26 May 2021

Journal Requirements:

>> We thank the editor for this comment. We have checked and updated the reference list accordingly. We fixed a mistake where the reference 

Campbell NL, Perkins AJ, Bradt P, Perk S, Wielage RC, Boustani MA, et al. Association of Anticholinergic Burden with Cognitive Impairment and Health Care Utilization Among a Diverse Ambulatory Older Adult Population. Pharmacotherapy. 2016;36(11):1123-31. Epub 2016/10/07. doi: 10.1002/phar.1843. PubMed PMID: 27711982; PubMed Central PMCID: PMCPMC5362375.

appeared in the reference list twice. This is also documented in the rebuttal letter. We did not cite any retracted manuscripts. 

Reviewers' comments:

Reviewer's Responses to Questions

Comments to the Author

1. If the authors have adequately addressed your comments raised in a previous round of review and you feel that this manuscript is now acceptable for publication, you may indicate that here to bypass the “Comments to the Author” section, enter your conflict of interest statement in the “Confidential to Editor” section, and submit your "Accept" recommendation.

Reviewer #1: All comments have been addressed

Reviewer #2: (No Response)

2. Is the manuscript technically sound, and do the data support the conclusions?

Reviewer #1: Yes

Reviewer #2: Yes

3. Has the statistical analysis been performed appropriately and rigorously? 

Reviewer #1: Yes

Reviewer #2: Yes

4. Have the authors made all data underlying the findings in their manuscript fully available?

Reviewer #1: Yes

Reviewer #2: Yes

5. Is the manuscript presented in an intelligible fashion and written in standard English?

Reviewer #1: Yes

Reviewer #2: Yes

6. Review Comments to the Author

Reviewer #1: Thank you for addressing the majority of the comments raised during the initial review. I believe the authors have substantially improved the manuscript. However, I do have a couple of additional questions that I would appreciate if the authors could address further regarding the study design.

1. I do appreciate that the authors have tried to generate a figure for the study design, however, it is not quite clarifying the cross sectional design to me. Particularly regarding the follow-up and dates that the morbidities and medications were identified on. For example, if the inclusion was met on February 1st 2016 for a patient, I assume the morbidities were assessed from January 1st 2004 through to January 31st 2016 (one day before the inclusion date), and medications were identified from January 1st 2015 through to January 31st 2016 (again one day before inclusion)? This is the level of detail in the figure that would be helpful to better understand the design.

>> The reviewer’s understanding of the time frames is correct. We have heeded the reviewer’s suggestion and created a new version of figure 1 where we clarified the timeframes in which morbidities and medications were assessed. In particular, we have now defined the day of inclusion in a footnote so that it is clear to which date the periods refer. 

2. On the point of morbidity and medication assessment prior to inclusion, this information (e.g., what were to morbidities and medications) is not summarized in the paper? I imagine this information could be helpful to understand differences in the patients with AB. Therefore, I wonder why this is excluded from the results since it appears to be assessed?

>> We thank the reviewer for this comment; In addition to the detailed information provided in Table S1, we have now added the following paragraph to summarize information on morbidity, medication as well as healthcare utilization in the study population, stratified by ACB status (ACB=1, ACB=2, ACB≥3 and persons without ACB) on page 7 lines 141-146

>> Page 7 lines 141-146: For example, compared to persons with lower or no ACB, persons with ACB≥3, had higher prevalences of psychiatric and behavioral, musculoskeletal as well as endocrine and metabolic diseases. They were prescribed medications from a higher number of different prescribers and had higher prevalences of cardiovascular therapy, analgesics and psychiatric medication. Moreover, persons with ACB≥3 were, on average, more frequently hospitalized, remained hospitalized for longer periods and had a higher prevalence of nursing home residency and obesity.

2. I am still unsure if this is truly a cross-sectional design as it does not appear to be a single snapshot of a patient at a given time-point. But instead there appears to be follow-up. For example, in the methods section and Figure 1, it states that patients were "followed until (i) death, (ii) start of hospitalisation in 2016 with a duration of >=90-days, (iii) end of continuous insurance period, or (iv) end of observation period, whichever occurred first". Based on this statement, the design still reads like a cohort study as you are following individual patients over time. With this in mind I wonder if it is a descriptive cohort study rather than a true cross-sectional design? Can the authors further clarify this. I think it would also help address the concerns from both reviewers regarding the potential for misclassification on the AB.

I understand the authors aim of the paper is only to assess the prevalence of AB in a younger population, which is of high interest. And while a cohort study that compares outcomes is not the aim, I do think further clarification on the follow-up and time of measurements is still warranted so that future work can replicate these findings and potentially conduct a cohort study to assess health outcomes.

>> We apologize that some parts of the description of the study design still created confusion. We fully understand that the expression “followed until…” was misleading here. We have now re-written this part of the methods section and hope it is clear now that this sentence should simply describe the observation period in 2016 that was used to assess the use of MAC. In a cross-sectional study based on primary data, one would ask patients, for example, about their MAC use in the past year. An assessment period of one year for MAC use seems reasonable to ensure that the assessment of MAC is robust regarding potential seasonal variations. In our study which was based on secondary data, this corresponds to defining an assessment period for MAC use of one year. Maybe this comparison is also helpful to explain the other measurement periods. In a primary data study, one would ask patients, for example, if they ever had a certain comorbidity (e.g. cancer). In our study based on secondary data, it is not possible to ask patients directly and pose the “ever” question. To compensate for that as much as possible, we used all the information available on this person in the database before 2016. This may also explain the measurement period we used for co-medication. In a primary data study, one would ask the patient about recent medication use (e.g. past 12 months). Analogously, we used an assessment period of one year to assess co-medication. We now expand on this also in the methods section and hope that this clarifies the study design.

>> Page 2 lines 60-62: For all included persons, the available (continuous) observation period in 2016 was used to assess the use of MAC. For persons with a hospitalization starting in 2016 and with a duration of ≥90 days, MAC use was only assessed until the start of this hospitalization (Figure 1).

>> Page 2 lines 65-71: The coding of morbidities was assessed any time prior to observation period (starting from 2004) through records of ≥1 ICD-10-GM inpatient or outpatient diagnoses or records of ≥1 codes of relevant operations, procedures or outpatient services as well as participation in disease management plans. This approach, i.e. taking into account all information on morbidity available for a person before 2016, aims to compensate for the fact that with secondary data, a person cannot be asked if he or she ever had a certain disease, as it would be done in a study based on primary data. Treatment with medication excluding MAC was assessed within 365 days before start of observation period (excluding start of observation period) based on records of ≥1 outpatient dispensations.

>> We also would like to point out that other cross-sectional studies based on claims data have used a similar approach in regards to the definition of observation periods, so for claims data analysis this is not an unusual approach (e.g. Simon et al, Pediatr Rheumatol Online J. 2020; 18: 43. (https://www.ncbi.nlm.nih.gov/pmc/articles/PMC7275412/) or Mapel et al BMC Health Serv Res. 2011; 11: 43. (https://www.ncbi.nlm.nih.gov/pmc/articles/PMC3050697/)

Reviewer #2: Thank you for the revisions. I think the study design is much clearer now and the explanatory figures included helpful.

The work done to explain that its really a cross-sectional prevalence (assessed over a one year period) is clarifying; but at the same time, I feel the implications of this requires some discussion or at least some further expansion in the limitations.

To the best of my knowledge (but this would be worth the authors discussing in any case) the time period over which the cumulative effects of anticholinergic burden accrue are unclear. The important point about high anticholinergic burden -- as the authors are well aware -- is not that people are on a number of medications, but rather than these medications together create an exposure that results in negative health outcomes like falls and dementia. But over what time period does this exposure need to occur? Does it matter if someone has a high level of anticholinergic burden for a 30 day period while a set of medications interact? Or does that cumulative exposure need to be over a 30 day period or 60 day period or for >1 year? How does the study design chosen illuminate the relationship between exposure and outcomes; or in another way, do we know that the exposures measured and reported here are clinically relevant?

To be clear, I don't think the authors need to actually answer all the questions above (and indeed, I believe some are presently unanswerable); but rather this is the context in which to discuss the strengths and limitations of their study design. I am unclear why reporting the results of a cross sectional measure of 'cumulative' (as the authors define is) is an important finding -- but i would welcome the authors defense of why this is a valid and important measure, through expansion of the discussion section.

>> We fully agree that the points the reviewer raised are very important and - even though partly unanswerable - should be mentioned in the discussion to sensitize the reader to uncertainties in the field. Therefore, we now expand on this in the discussion in a separate paragraph (page 15, lines 244-253). With respect to the ‘cumulative” AB: We hope our answer regarding the study design we provided in response to reviewer 1 explains to a certain extent why we think this measure is not contradictory to a cross-sectional study. Also in a cross-sectional study based on primary data, one would ask a patients about MAC use e.g. in the past year, and then there are different options how to categorize / classify this information to describe the prevalence of AB. Apart from describing the persons regarding the highest category of AB reached during the study period, we calculated this additional measure because – unlike the aforementioned measure - it allowed us to assess the proportion of AB attributable to a certain class of MAC or to a certain physician specialty. We think this additional measure is helpful as it provides insights into the medication and sectors of care that contribute to the AB in the population. 

>> Page 15 lines 244-253: Our study showed that there are persons with an AB considered to be clinically relevant in all age groups. This demonstrates the need to conduct studies on potentially harmful effects not only in older adults but also in children, adolescents, and the entire adult population. However, it has to be kept in mind that there are a lot of unanswered questions in regards to how AB can cause or contribute to clinically relevant adverse effects. For example, the time period over which the cumulative effects of anticholinergic burden may accrue and possibly produce harms are unclear. Also the role of type and dosage of single MACs and their overlap are not well understood. When planning a study on the risk of AB, this means that classifying persons as exposed or unexposed bears a high level of uncertainty, so robustness of findings would need to be assessed by comprehensive sensitivity analyses. Also in many other regards, studies on the risk of outcomes associated with AB are challenging, e.g. regarding issues such as confounding by indication, unmeasured confounding and time-varying exposure. 

At the very least, I would think a clear limitation to the design should be added to state that longitudinal assessment of anticholinergic burden that accounts for dose is what would be needed to understand the link between anticholinergic burden exposure and negative health outcomes -- particularly in the younger population who may have higher than expected anticholinergic burden but to my knowledge have never been shown to develop the negative health outcomes observed in older adults.

I recognize that the authors have inserted this limitation related to the above: 'It should be kept in mind, however, that this measure might not be suitable for studies assessing the risk of outcomes associated with AB where a precise classification of exposed and unexposed time windows is important.' However in my mind, that is the important goal of this type of work. So discussing/addressing that limitation more fulsomely would be useful.

>> We apologize that we had not sufficiently addressed this limitation yet. We now expand on this in the limitations section and – as also mentioned above –have added a paragraph addressing the open questions in this regard. 

>> Page 16 lines 274-279: Nonetheless, this approach is not equivalent to other studies that had more information on dosage and used more sophisticated methods to take it into account. Finally, in our study we have not assessed AB in a longitudinal manner, which – in view of the aforementioned unanswered questions about clinically relevant AB levels – would be essential in a subsequent risk study to understand the potential link between AB exposure and negative health outcomes. Such risk studies are particularly needed in the younger population where it is even less clear if such a link exists at all.

7. PLOS authors have the option to publish the peer review history of their article (what does this mean?). If published, this will include your full peer review and any attached files.

Do you want your identity to be public for this peer review? For information about this choice, including consent withdrawal, please see our Privacy Policy.

Reviewer #1: No

Reviewer #2: No

---

## [Editor Report · Decision Letter 2]

3 Jun 2021

Anticholinergic burden: first comprehensive analysis using claims data shows large variation by age and sex

PONE-D-20-37319R2

Dear Dr. Reinold,

We’re pleased to inform you that your manuscript has been judged scientifically suitable for publication and will be formally accepted for publication once it meets all outstanding technical requirements.

Kind regards,

Andrea Gruneir

Academic Editor

PLOS ONE
---

## [Editor Report · Acceptance letter]

7 Jun 2021

PONE-D-20-37319R2 

Anticholinergic burden: first comprehensive analysis using claims data shows large variation by age and sex 

Dear Dr. Reinold:

I'm pleased to inform you that your manuscript has been deemed suitable for publication in PLOS ONE. Congratulations! Your manuscript is now with our production department. 

Kind regards, 

on behalf of

Dr. Andrea Gruneir 

Academic Editor

PLOS ONE